# In Vitro Human Joint Models Combining Advanced 3D Cell Culture and Cutting-Edge 3D Bioprinting Technologies

**DOI:** 10.3390/cells10030596

**Published:** 2021-03-08

**Authors:** Christian Jorgensen, Matthieu Simon

**Affiliations:** 1UFR de Médecine, Université MONTPELLIER 1, 34000 Montpellier, France; 2IRMB, CARTIGEN, CHU de Montpellier, 34090 Montpellier, France; matthieusimon.work@gmail.com

**Keywords:** musculoskeletal progenitor/stromal cells, organoids, microspheres

## Abstract

Joint-on-a-chip is a new technology able to replicate the joint functions into microscale systems close to pathophysiological conditions. Recent advances in 3D printing techniques allow the precise control of the architecture of the cellular compartments (including chondrocytes, stromal cells, osteocytes and synoviocytes). These tools integrate fluid circulation, the delivery of growth factors, physical stimulation including oxygen level, external pressure, and mobility. All of these structures must be able to mimic the specific functions of the diarthrodial joint: mobility, biomechanical aspects and cellular interactions. All the elements must be grouped together in space and reorganized in a manner close to the joint organ. This will allow the study of rheumatic disease physiopathology, the development of biomarkers and the screening of new drugs.

## 1. Introduction

Osteoarthrosis (OA) is a chronic degenerative disease of diarthrodial joints affecting people over forty resulting in chronic joint pain and progressive loss of mobility. In 2019 OA affected about 7% of the world’s population [1]. So far, there is no therapy available that effectively stops the structural degradation of cartilage and bone or is able to successfully reverse this joint deterioration [2]. OA leads to chronic pain, and severe restriction in mobility inducing loss in quality of life. This statement emphasizes the need to develop new therapeutics in order to improve the current treatment which consists only of killing the patient’s pain. Nowadays two different models are commonly used to better understand OA physiopathology, to develop biomarkers and assay new drugs. These models are animals and in vitro models [3]. Both models have advantages and disadvantages (Table 1). In vitro models and cell cultures are easy to set up, as a large number of cells can be produced allowing high throughput drug screening applications. However, due to their simplicity these models cannot properly replicate the complexity of an organ like a diarthrodial joint, such as intracellular signaling, fluid forces or the influence of tissue interconnexion. On the other hand, animal models like the DMM model (disease induced by menisectomy) or CIOA (collagenase-induced osteoarthritis) bring this missing organ complexity and have largely contributed to a better understanding of the function of the joint. Unfortunately, the use of animals and more particularly large ones which better mimic human articulation is limited because of ethical concerns and the obvious practicality [4]. In a recent review Cope et al. discussed all the advantages and disadvantages of in vivo and ex vivo models to study OA [5]. They finally concluded that at this time no suitable model accurately reflected the natural human OA particularly at the early stages of the disease when a preventive treatment could be initiated to slow or even reverse the progression of the disease.

The complexity of OA was highlighted only recently [6]. Indeed, OA was initially considered as a cartilage disease but a few studies over the last decade have demonstrated that OA results from cellular changes and biomechanical stresses coming from the whole articulation [7,8]. Joint tissue is a complex tissue and it is difficult to replicate the microenvironments in vitro in order to mimic joint pathologies. The development of new biological human joint models in order to predict the efficacy and toxicity of new treatments in rheumatological diseases is therefore essential but challenging.

New biological models of human articulation are currently emerging. They combine advanced 3D cell culture and cutting-edge technologies. We can expect that these new biological models will in the near future lower or even eliminate the use of animal models as preclinical models of osteoarticular pathologies (Figure 1). First, organ-on-a-chip engineering is a new technology able to replicate organ functions into microscale systems close to pathophysiological conditions [9]. On the other hand, recent advances in 3D printing techniques allow the precise control of the architecture of cellular compartments. Thanks to perfusion bioreactors for 3D constructs [10], those powerful tools can integrate fluid circulation, the delivery of growth factors and cytokines, and physical stimulation including oxygen level, external pressure and mobility. We will summarize here our current knowledge on these emerging techniques and we will review the latest scientific and technological advances in the field of joint biomimetic systems.

## 2. Organoids

Organoids are made of stem cells or progenitor cells which can self-organize in three dimensions leading to a simplified version of an organ [11]. They offer the opportunity to understand complex biological systems such as joint tissues in a physiologically relevant context where two-dimensional models have not proven as successful. The development of the joint organoid is highly dependent on the cell environment.

The extracellular matrix (ECM) of articular cartilage is a complex environment whose composition and structure allow repeated loading and physical stimulation in the mobile organ. Cartilage ECM is composed of a fluid phase and a solid phase primarily comprising fibrillar and nonfibrillar collagens (mainly collagen types II and I), proteoglycans (mostly aggrecan and lubricin) and hyaluronic acid. The physiological compression of articular cartilage induces a complex mechanical environment that is characterized by stresses, strains, osmotic and hydrostatic pressures, interstitial fluid flow, and electrokinetic effects varying in time and space. A first work by Broeren et al. has shown that it is possible to mix human synovial microblasts with human monocytes to form a micromass in a drop of Matrigel [12,13]. These elements contained 50,000 synovial cells from subjects with rheumatoid arthritis mixed with one million monocytes. The mature micromass was then inoculated on a 24-well plate before being stimulated for three weeks with cytokines, either TNF alpha or TGF beta 1. The inflammatory cytokine induced phenotypes characteristic of polyarthritis: fibrosis, hyperplasia and synovial proliferation, the induction of inflammatory cytokines. On the other hand, TGF beta induced a rather fibroblastic phenotype closer to osteoarthritis disease. These elements show that indeed the combination of the two cell types, synovial and monocytes, can enable the study and the influence of cytokines on these cellular elements.

Indeed, Han et al. have shown that it is possible to organize micromasses combining chondrocytes and synovial cells [14]. They obtained the most relevant ratio with one synovial cell for three chondrocytes and made it possible to increase the expression of chondrogenic markers in the micromass such as type II collagen, the expression of proteoglycans or even Sox9 transcription factors. The combination of the two optimized the chondroinductive effect. We ourselves have shown a reduction in inflammatory characterizations of chrondrocytes in the presence of synovial cells or in the presence of mesenchymal cells [15]. Thus, cytokines like IL-6 and IL-8 or chemokines were reduced in these coculture models. In the same way, it was possible to mix chondrocytes, synovial cells and monocytes in three dimensions. This time, a more complex model made it possible to observe the impact on chondrocyte cells, in particular, the increase in apoptosis, the induction of enzymes such as metalloproteases and the reduction of constituents of the extracellular matrix. This model allowed to some extent the replication of articular compounds. Administration of celecoxib significantly reduced the expression of metalloproteases, indirectly validating this model [16].

The subchondral bone component is critical in the physiopathology of OA. Thus, the osteochondral progenitor microspheres were stimulated for 3 weeks by TGFb and then sequentially mixed with an osteogenic medium [17]. This made it possible to obtain mixed microspheres with a bone and cartilage structure at the periphery. Once implanted, these micromasses behaved like a functional tissue recruiting hematopoietic stem cells to the recipient and reconstituting hematopoiesis. These functional microspheres could then be subjected to physical constraints. This is what we achieved in the laboratory with a device that allowed the increase of pressure and the measurement of the Young’s modulus and the Poisson ratio as a function of the constituents of the extracellular matrix and on the chondrocyte phenotype (N. Petitjean et al., in press).

## 3. Joint-on-a-Chip

The last decades have seen a strong evolution in micro fabrication and soft lithography techniques [18] which have led to the development of increasingly complex biological systems able to reproduce the functionality of an organ. Indeed, “organ-on-a-chip” is a miniaturized system where cells are cultured in microfluidic channels which can be compartmentalized to accommodate different types of cells (cell coculturing). Microfluidics maintain cells alive by letting the culture medium flow for several weeks providing conditions close to those existing in vivo (pH, flow, pressure and nutrient). The researcher can also tune specifically one of these conditions or test the effect of different drugs on the cell behavior. The organ-on-a-chip system has recently been allowed to replicate the functions of different organs like heart, liver, lung, skin, gut and brain [19,20,21].

Concerning the joints, cartilage-on-a-chip systems have also recently emerged [22,23,24]. Cartilage is an avascular and non-innervated tissue and comprises only one type of cells: chondrocyte. Most of the stimuli received by chondrocytes are mechanical. In order to better understand this mechanical influence on cartilage, Paggi et al. developed a microdevice able to induce a controlled pressure through actuation chambers directly on the chondrocyte-laden hydrogel [22]. They showed that the mechanical stimulation enhanced pericellular matrix formation and glycosaminoglycans (GAGs) content.

Another similar example is Occhetta’s work which allowed a 3D microconstruct to be deposited in a device that can induce pressure in a central chamber [23]. This technique combines soft lithography, organoid and organ-on-a-chip technologies and makes possible the analysis in chondrocyte cells of the expression of genes necessary for the extracellular matrix, such as proteoglycans or collagen II, and also the pathways involved in tissue regeneration including Frizzled related protein (FRZ-B) and Dickkopf-related protein 1 (DKK1) which are involved in the regulation of chondrocyte maturation and long bone development. In this device, the authors were able to analyze the effect of validated treatments for osteoarthritis such as a nonsteroidal anti-inflammatory drug, dexamethasone, hyaluronic acid and the IL-1b antagonist and showed a beneficial effect on the expression of metalloproteases validating the device.

However, as explained above, OA is not only a cartilage disease. This statement highlights one limitation of current organ-on-a-chip systems: these devices are not yet able to fully duplicate the complexity of the organ studied. Indeed, joint-on-a-chip involves reassembling all the cell types involved in a mini-organ: chondrocytes, osteoblasts, immune cells, synovial cells and adipose tissue. All of these structures must be able to mimic the specific functions of the joint: mobility, biomechanical aspects and cellular interactions. All the elements must be grouped together in space and reorganized in a manner close to the joint organ. The joint is submitted to various forces as well as mobility. The biomechanical constraints on the different cellular compartments are different. Thus, synovial cells are subjected to stresses and strains related to fluids and the chondrocytes are subjected to compression and stretching phenomena. This is the case with the tendons. Adipose tissue is subjected to stresses associated with fluids and compression elements, as is the case with meniscal fibroblast cells.

Thus, in order to mimic an articulation in vitro, the biomechanical aspect must be taken into account. These mechanical stimulations can be provided by mini-bioreactors or microfluidic models available today.

## 4. Devices Mimicking the Joint Microenvironment

Micro-bioreactors have been proposed in these technological aspects: the device is a chamber with a lower circulation and an upper circulation. These scaled-down models of laboratory bioreactors can easily be multiplied to perform high-throughput studies such as drug screening. Within the device are combined the various constituent elements including the cells of cartilage, bone, synovium and endothelium [25]. The combination of these different elements allows the study of the circulation of fluids containing cytokines. Thus, in this model, the administration of IL-1b makes it possible to mimic the OA phenotype. In fact, in the upper part of the device, the cartilage tissue then expresses matrix metalloproteinases, MMP-2, MMP-3 and ADAMs-4, and at the same time, reduces the expression of collagen II and proteoglycans. The simultaneous administration of an anti-inflammatory drug such as 10 µmol celecoxib prevents this deleterious effect and again validates the method. These devices allow the inclusion of progenitors derived from induced pluripotent stem cells (IPS) allowing the study of the development end regenerative process after injury [26].

## 5. 3D Bioprinting

Three-dimensional bioprinting can overcome the organ-on-a-chip size limitation to mimic the complexity of joint tissue. Historically 3D printing began in the early 2000s, with the use of heated resin and was used for rapid prototyping. Quickly, biomedical applications have emerged with, for example, 3D bioprinting which consists of the creation of living tissue using the additive manufacturing technology of 3D printing [27,28]. Three -dimensional bioprinting can transform 3D virtual models created with computer-aided design (CAD) into physical objects through the deposit of a hydrogel containing cells, also called bioink, in a layer-by-layer manner. So, depending on the desired 3D bioprinting application three components will be highly tunable: the 3D printer corresponding to a specific printing technique, the nature of the hydrogel and the cells. Extensive literature concerning bioprinting has emerged in the last few years [29,30,31,32,33,34], in this review we will focus on examples concerning tissues present in the joint.

Extrusion is the most widespread method for fabricating cartilage tissues [35,36]. Indeed, chondrocytes grow and proliferate in aqueous hydrogel, such as collagen and its derivative, chitosan, alginate, agarose. In this system the bioink is pushed through a nozzle, in a layer-by-layer manner. The obtained construct usually displays high cell viability (>95%). However, hydrogels lack the mechanical strength found in the natural tissue. In order to increase the hydrogel’s mechanical properties, hydrogels can be combined with synthetic scaffold made of plastic polymer (PLA or PCL) printed using stereolithography or fuse deposit modeling 3D printing. For example, Bahececioglu et al. 3D printed a PCL scaffold mimicking a meniscus and impregnated it with agarose in the inner region and with gelatin methacrylate (GelMA) in the outer region [37]. After seeding the construct with porcine fibrochondrocytes and after 8 weeks of incubation with or without dynamic stimulation, they showed that agarose enhanced GAG production while GelMA enhanced collagen production corresponding, respectively, to a hyalin cartilage at the inner portion and fibrocartilage at the outer portion. This publication highlights the importance of the bioink composition and the role of the mechanical stimulation in the cell fate.

Indeed, a large number of studies aim at developing new suitable biomaterials for 3D bioprinting in general. The main objective of these studies is to design and synthesize new bioinks derived from natural or fully synthetic polymers which perfectly mimic the desired cellular environment in order to positively influence cell development towards a cell type in a perfectly controlled and reproducible manner. Furthermore, these modifications allow the chemical linking of bioactive molecules, such as growth factors, nanoparticles, micro carriers, peptides that will give additional properties to the bioink, for example, hydroxyapatite grafting could increase gel stiffness and improve cell differentiation in osteoblasts [38]. In addition, the gelation process of these new bioinks must be perfectly controlled to be able to encapsulate the cells and then to be 3D printed. Finally, these bioinks must also display good swelling, thermal stability, particularly in an aqueous media at 37 °C, and most importantly exhibit the lowest possible cytotoxicity. A good example for increasing the thermal stability of a hydrogel is the introduction of a methacrylate group into biopolymers such as gelatin or collagen, this chemical modification allows the control of the gelation process of the gelatin after UV exposure and in the presence of a photoinitiator. Following this modification, the GelMA hydrogels are gelled and stable at 37 °C, in contrast to the natural biopolymer. However, acrylate polymerization has a major disadvantage which is the generation of reactive oxygen species (ROS). This was shown by Roberts et al. who compared the impact of acrylate-based PEG or thiolene PEG hydrogels on cartilage development of bovine chondrocytes [39]. They showed that the acrylate-based hydrogel induced elevated intracellular ROS level and favored hypertrophic cartilage development while the thiol norbornene system favored hyaline cartilage development and displayed a low level of ROS. This statement emphasizes the need to further develop new innovative bioinks.

In 2015, Gao et al. used inkjet bioprinting methods, which relied on the ejection of a small amount of bioink with low viscosity, to seed and simultaneously photocrosslink a scaffold made of polyethylene glycol (PEG) and GelMA with human MSCs [40]. They demonstrated an improvement of mechanical properties and osteogenic and chondrogenic differentiation, suggesting its promising potential for usage in bone and cartilage tissue engineering.

Those studies are interesting since they demonstrate well the increasing complexity in the field of cartilage or bone 3D bioprinting. Indeed, the development of new bioinks, the fusion of different 3D printing methods, the development of more and more complex devices able to mimic natural environmental stimuli and the interconnexion between organoid, organ-on-a-chip and 3D printing techniques is in constant progress [41]. Thus, we can expect in the near future, the development of new humanized models based on 3D printing able to replicate the whole joint and not only cartilage or bone tissues.

## 6. Conclusions

In conclusion, the implementation of micro-bioreactors in devices or the development of organoids-on-a-chip now make it possible to validate the joint phenotype either as a model of osteoarthritis or as a model of inflammatory diseases such as rheumatoid arthritis. These organoids can be subjected to oxygen differentials comparable to what we observe in human pathology and to mechanical stresses either by pressure on the organoid itself or by microfluidic circulation. These models, whether they are micro-bioreactors or organoids, have made it possible to validate the use of drugs used in human pathology such as glucocorticoids, IL-1b inhibitors or hyaluronic acid. These new models are validated and make it possible to propose alternatives to experimental animal models. This is in line with the reduction of preclinical models in the identification of new treatments or new regeneration strategies in joint pathology.

## Figures and Tables

**Figure 1 cells-10-00596-f001:**
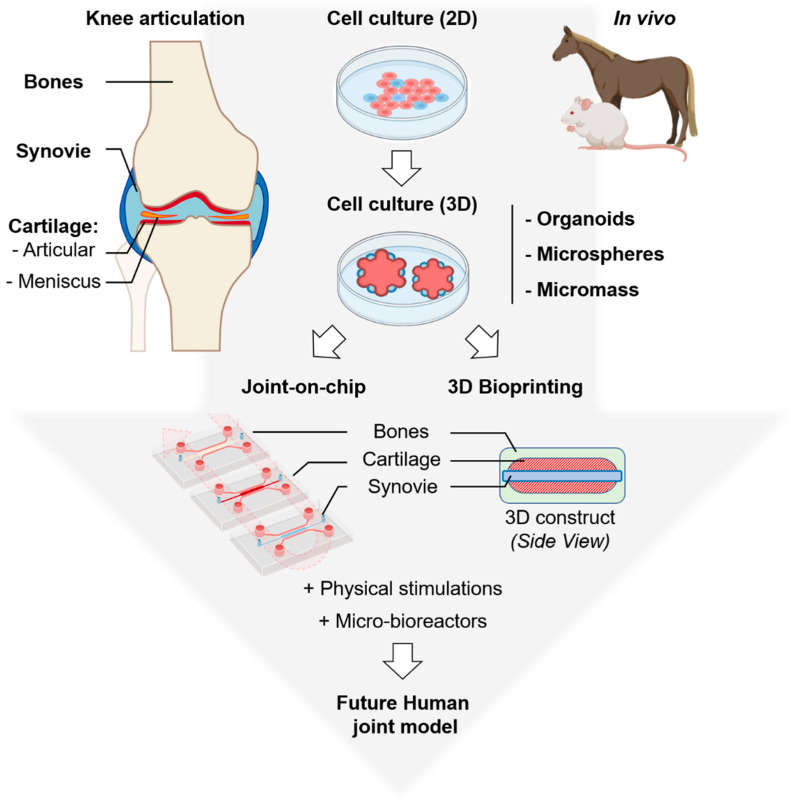
Overview of the biological models commonly used: (**top**) animals, 2D and 3D cell cultures and (**bottom**) emerging: joint-on-a-chip and 3D bioprinting to study human articulation and its related diseases. Organoids are complex clusters of organ-specific cells, microspheres refer to shape-specific, macroporous scaffolds seeded with cells and micromass corresponds to high-density 3D cell cultures with multilayered organization. Top left, a schematic representation of the whole knee articulation, bones are in light brown, synovial cavity in light blue, articular cartilage in red and meniscus in orange, this color code was used for all the figures. Parts of the figure were created using biorender.com.

**Table 1 cells-10-00596-t001:** Pros and cons of the different joint models.

Models	Benefits	Limitations
Animals	Biological relevance	Ethical concerns
		Pharmacology different between species
		Costs, infrastructure and skills required
Cell culture	Easy to manipulate	No fluid forces
	Simple	Change in phenotype
Organoid	Improve cell function and gene expression	Immaturity of cells
	Relevant microenvironment	
	Multiple cell types	
3D bioprinting	Anatomy of the organ	Difficult to handle
	vascularization	
	Size	
Joint-on-a-chip	Chip interconnexion	Anatomy of the organ Size
	Cell coculturing	
	Air liquid interface	

## Data Availability

No new data were created or analyzed in this study. Data sharing is not applicable to this article.

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
