# Peer review of "In Vitro Human Joint Models Combining Advanced 3D Cell Culture and Cutting-Edge 3D Bioprinting Technologies"

_cells, 2021, doi:10.3390/cells10030596_

Round 1
Reviewer 1 Report
This paper reviews the latest advances in 3D cell culture techniques as promising alternatives to the currently used animal models and 2D tissue culture. While the topic is timely and relevant, I would suggest that the authors do an in depth reference search to ensure that they are covering each area in comprehensive manner consistent with a review paper. Please find additional suggestions for improvement below.
The title does not accurately reflect the topics covered in the review. I would suggest adding more critical analysis of each of the new techniques described in terms of advantages and limitations along with more details on the potential for each of the techniques (organoids, 3D printed constructs, lab on a chip model). A more appropriate title might also include the potential of these techniques in addition to their challenges.
It would be helpful in the abstract to clarify what type of joints are being referred to. It would also be good to include what sort of cells need to be present in these models. What types of structures are present in cartilage? This information should be included in the abstract.
Line 27 – the paper refers to 2 models and more 2 models are discussed in the paper so I would revise this sentence.
Line 35-36 – it would be helpful to discuss the challenges associated with treating with osteoarthritis and the limitations of the current systems.
Line 43 – this sentence needs a citation.
Line 51 – this sentence is awkward and should be rephrased.
Line 58 - please define what is meant by precise control.
In Section 2, it would be helpful to define what is meant by an organoid, microsphere, and micromass. I’m not sure that the term microsphere is being used in the correct context.
Line 71 needs a citation.
Line 78 – it would be helpful to further detail the composition of the ECM.
Line 86 – what was the source of the monocytes? Human? Same line – what is “whole” referring to?
Which version of TGF beta is inflammatory?
Line 89 contains an incomplete sentence – this should be corrected.
I would use a consistent term - organoids or micromasses - or create a table with the definitions.
How do the cell ratios used compared to the concentrations found in vivo? That would be helpful to include. It would be helpful to elaborate how these co-culture models mimic the structures found in vivo.
In Section 3, the use of lab-on-a-chip systems has been around longer than the last few years. I would suggest expanding this section to include more studies as well as including more discussion on what would a multicompartment chip look like for these systems. It would also be helpful to include figures to help visualize the devices.
Line 127 – I would suggest including a more recent review as well.
Line 140- what is being referred to by FRZ-B and DKK1?
Section 4 should define what a microreactor is for the benefit of the reader.
5.3’s heading should be 3D bioprinting and I would suggest adding additional references to the Introduction paragraph. I would also suggest finding additional studies that 3D printed cartilage models to expand this section. Defining what extrusion bioprinting would also be helpful for readers. For existing studies, I would add detail on what types of cells were printed and the functionality of these constructs.
Line 189 – please quantify what high viability is.
What would be the desired mechanical properties for such constructs?
Line 221- what system were these effects in observed in?
I would expand the conclusions section to include the potential for future work in this area.
All abbreviations should be written out the first time they are used.
Author Response
Referee 1
This paper reviews the latest advances in 3D cell culture techniques as promising alternatives to the currently used animal models and 2D tissue culture. While the topic is timely and relevant, I would suggest that the authors do an in-depth reference search to ensure that they are covering each area in comprehensive manner consistent with a review paper. Please find additional suggestions for improvement below.
According to referee 1 and 2 additional references were added to cover each area.
The title does not accurately reflect the topics covered in the review. I would suggest adding more critical analysis of each of the new techniques described in terms of advantages and limitations along with more details on the potential for each of the techniques (organoids, 3D printed constructs, lab on a chip model). A more appropriate title might also include the potential of these techniques in addition to their challenges.
The title of the review was changed to “Emergence of in vitro human joint models combining advanced 3D cell culture and cutting-edge 3D bioprinting technologies . “
It would be helpful in the abstract to clarify what type of joints are being referred to. It would also be good to include what sort of cells need to be present in these models. What types of structures are present in cartilage? This information should be included in the abstract.
Abstract: Joint on chips are new technologies able to recapitulate the joint functions into microscale systems close to pathophysiological conditions. Recent advances in 3D printing techniques allow precise control of the architecture of cellular compartments (including chondrocytes, stromal cells, osteocytes, synoviocytes). These tools integrate fluid circulation, the delivery of growth factors, physical stimulation including oxygen level, external pressure, and mobility. All of these structures must be able to mimic the specific functions of the diarthrodial joint: mobility, biomechanical aspects and cellular interactions. All the elements must be grouped together in space and reorganized in a manner close to the joint organ. This will allow the study of rheumatic diseases physiopathology, develop biomarkers and screen new drugs.
Line 27 – the paper refers to 2 models and more 2 models are discussed in the paper so I would revise this sentence.
Agree. This sentence refers to the 2 models commonly used today and in the past (animals and 2D cultures) and not the emerging ones that are discussed later.
Line 35-36 – it would be helpful to discuss the challenges associated with treating with osteoarthritis and the limitations of the current systems.
Agree. We added limitation of the current models are discussed Lines 29-41.
Line 43 – this sentence needs a citation.
A citation was added:
Carlesso, L.C.; Neogi, T. Understanding the Complexity of Pain in Osteoarthritis Through the Use of Pain Phenotyping: Current Evidence. Curr Treat Options in Rheum 2020, 6, 75–86, doi:10.1007/s40674-020-00144-z
Line 51 – this sentence is awkward and should be rephrased.
This sentence was rephrased:
“New biological models of human articulation are currently emerging. They combine advanced 3D cell culture and cutting-edge technologies. We can expect that these new biological models will in the near future lower or even eliminate the use of animal models as preclinical models of osteo-articular pathologies.”
Line 58 - please define what is meant by precise control.
“Precise control” is bringing by the use of a 3D printer. Explication is in the 3D bioprinting section.
“Quickly, biomedical applications have emerged with, for example, 3D bioprinting which consists in the creation of living tissues using the additive manufacturing technology of 3D printing. 3D bioprinting can transform 3D virtual models created with computer-aided design (CAD) into physical objects through the deposit of a hydrogel containing cells, also called bioink, in a layer-by-layer manner.”
In Section 2, it would be helpful to define what is meant by an organoid, microsphere, and micromass. I’m not sure that the term microsphere is being used in the correct context.
For clarity, microsphere, organoid and micromass definitions were added in the legend of the Figure 1:
“Organoids are complex clusters of organ-specific cells, microspheres refer to shape-specific, macroporous scaffolds seeded with cells and micromass corresponds to high-density 3D cell cultures with multilayered organization.”
Line 71 needs a citation.
A citation was added:
Kim, J.; Koo, B.-K.; Knoblich, J.A. Human Organoids: Model Systems for Human Biology and Medicine. Nat Rev Mol Cell Biol 2020, 21, 571–584, doi:10.1038/s41580-020-0259-3
Line 78 – it would be helpful to further detail the composition of the ECM.
We describe the cartilage ECM is composed of a fluid phase and a solid phase primarily comprising fibrillar and non-fibrillar collagens (mainly collagen types II and I), proteoglycans (mostly aggrecan and lubricin) and hyaluronic acid.
Line 86 – what was the source of the monocytes? Human? Same line – what is “whole” referring to?
Monocytes were from human and “whole” is referring to mature micromass.
Thus, the sentence was modified:
“A first work by Broeren et al. has shown that it is possible to mix human synovial micro-blasts with human monocytes to form a micromass in a drop of Matrigel [11]. These elements will contain 50,000 synovial cells from subjects with rheumatoid arthritis mixed with one million monocytes. Mature micromass is then inoculated on a 24-well plate before being stimulated for three weeks with cytokines, either TNF alpha or TGF beta 1.”
Which version of TGF beta is inflammatory?
TGF beta 1
Line 89 contains an incomplete sentence – this should be corrected.
Sentence was corrected:
“The inflammatory cytokine will induce phenotypes characteristic of polyarthritis: fibrosis, hyperplasia and synovial proliferation.”
I would use a consistent term - organoids or micromasses - or create a table with the definitions.
For clarity, microspheres, organoid and micromass definitions were added in the legend of the figure 1.
How do the cell ratios used compared to the concentrations found in vivo? That would be helpful to include. It would be helpful to elaborate how these co-culture models mimic the structures found in vivo.
In vivo , the tissue are in contact but separated. The epiphysial bone is covered by cartilage then virtual cavity with meniscus (fibrocartilage) and synovial tissue in the recesses. Thus it is impossible to identify specific cell ratio in vivo. The ratio presented are discussed in the following publication that was added:
Broeren, M.G.A.; de Vries, M.; Bennink, M.B.; Arntz, O.J.; van Lent, P.L.E.M.; van der Kraan, P.M.; van den Berg, W.B.; van den Hoogen, F.H.J.; Koenders, M.I.; van de Loo, F.A.J. Suppression of the Inflammatory Response by Disease-Inducible Interleukin-10 Gene Therapy in a Three-Dimensional Micromass Model of the Human Synovial Membrane. Arthritis Res Ther 2016, 18, 186, doi:10.1186/s13075-016-1083-1
In Section 3, the use of lab-on-a-chip systems has been around longer than the last few years. I would suggest expanding this section to include more studies as well as including more discussion on what would a multicompartment chip look like for these systems. It would also be helpful to include figures to help visualize the devices.
Agree. The sentence was modified as following :
“The last decades have seen a strong evolution in micro fabrication and soft lithography techniques.”
Line 127 – I would suggest including a more recent review as well.
More recent reviews were added as suggesting by referee 1.
Line 140- what is being referred to by FRZ-B and DKK1?
FRZ-B, DKK1 and other abbreviations were written out the first time they are used.
Section 4 should define what a microreactor is for the benefit of the reader.
Microreactor was quickly defined:
“The device is a chamber with a lower circulation and an upper circulation. These scaled-down models of laboratory bioreactors can easily be multiplied to perform high-throughput studies such as drug screening.”
5.3’s heading should be 3D bioprinting and I would suggest adding additional references to the Introduction paragraph. I would also suggest finding additional studies that 3D printed cartilage models to expand this section. Defining what extrusion bioprinting would also be helpful for readers. For existing studies, I would add detail on what types of cells were printed and the functionality of these constructs.
Section title was corrected, extrusion bioprinting was defined and citations concerning cartilage and 3D bioprinting were added:
“Extrusion is the most widespread method to fabricate cartilage tissues [35,36].”
“3D bioprinting which consists in the creation of living tissues using the additive manufacturing technology of 3D printing. 3D bioprinting can transform 3D virtual models created with computer-aided design (CAD) into physical objects through the deposit of a hydrogel containing cells, also called bioink, in a layer-by-layer manner.”
Line 189 – please quantify what high viability is.
This was quantified:
The obtained construct usually displays high cell viability (> 95%).
What would be the desired mechanical properties for such constructs?
“the mechanical strength corresponding to the natural tissue”
Line 221- what system were these effects in observed in?
The nature of cells used in this study was added:
“This was shown by Roberts et al. who compared the impact of acrylate-based PEG or thiolene PEG hydrogels on cartilage development of bovine chondrocytes.”
Reviewer 2 Report
The paper gives an overview of the topic and describes several approaches for developing joint chip models. I think that discussing the applications of these models more clearly and what the barriers to their uptake are would be a useful addition to the paper, beyond the nice examples already provided. How far away are we from using the models in drug development and what do we need to accelerate this? Could the authors also consider OA in animals - particularly horses and dogs - and whether joint chip development could be used to develop possible treatments or further our understanding of the disease in other species? Pregenerate (https://www.pregenerate.net/about) are developing cartilage chip models using animal cells, and the authors may want to be aware of their research.
Figure 1 is very clear and beautifully presented, I am not sure that the animals add anything as you do not mention these in the legend. I really like the colour coding.
The section on Joint on chip is very good, I think that you should try and mirror this for the section on organoids, which is not as clear.
I also have these specific, minor issues that I think you could address:
Line 22: I think it would be more relevant to refer to more recent data regarding disease prevalence (eg Hunter et al. 2020 Osteoarthritis in 2020 and beyond: a Lancet Commission. The Lancet).
Lines 31-33 - I agree that 2D cultures are not sophisticated enough to model the joint, but this explanation is too generic and I think that you need to refer to the architecture and cell types of the joint, not just refer to "an organ", since your review is focused on the joint specifically, not organs in general.
Line 33-35: This seems misleading, animals can contribute to better understanding of animal biology, but can only lend assumptions to the understanding of human biology! We need to study humans to truly understand human biology.
Table 1 - in the first row, you need to explain what you mean by "interspecies" and there are also other issues associated with animals (costs, infrastructure, skills required, etc). For Cell culture - I do not think that you are correct- do you mean that cell cultures cannot recreate intercellular communication?
Line 48- I think you need to insert the term "in vitro" after "microenvironments" for clarity
Line 48 - you need to clarify what you mean by "humanised models" -are you referring to in vitro/ex vivo models based on human cells or in vivo animals that are transgenic for human genes?
Line 57-59 is not clear- as I understand it, 3D printing does not always/automatically integrate fluid circulation etc, but 3D printed structures can be incorporated into dynamic models to produce the features that you describe.
Line 84-91 This is an interesting example, did the authors of the paper (Broeren et al) create models with non-arthritic cells as a comparator? It is not clear what this research added to the field- the need for continued stimulation of cultures to maintain the pro-inflammatory profile, or the fact the these cells could be cultured together as a disease model?
Line 105-106 regarding the effects of celecoxib needs a reference.
Line 110 - can you define where the microspheres were implanted- I assume this is an animal model but that needs to be clarified.
Line 114 It is Young's modulus (not Yong's)
Line 115-116, it would help to describe more about this model that you are developing - what results did you achieve when you altered the pressure and how does this mirror the disease? Can you comment on the possible development of this model for drug screening/understanding disease progression?
Line 140 Define the abbreviations FRZ-B and DKK1 please
Line 141 this sentence does not make sense
"different treatments already validated in human disease as osteoarthritis"
Line 157 I am not sure what you mean by this:
"In order to reconstruct an articulation on a chip, we will rely on biomaterials, 3D culture and mini-bioreactors or microfluidic models available today." The English is not quite clear.
Line 167 Define MMP and ADAM please.
Line 171 Define IPS
Line 174 I disagree with the statement "3D bioprinting can overcome the organ-on-chip limitation to mimic the complexity of joint tissue." as I think you are oversimplifying both techniques! I think that 3D bioprinting offers some advantages, but it cannot substitute for OOC technologies and we need to consider combining this with other approaches, such as OOC, then you can start to build a more complex model with flow, vasculature, immune cell circulation etc., but I do not agree that these techniques are mutually exclusive, as you seem to be suggesting.
Line 201- could you clarify this please- do you mean for joints specifically or in general?
Line 206: I do not know what you mean by "doping agents" and perhaps you could explain or rephrase?
Author Response
The paper gives an overview of the topic and describes several approaches for developing joint chip models. I think that discussing the applications of these models more clearly and what the barriers to their uptake are would be a useful addition to the paper, beyond the nice examples already provided. How far away are we from using the models in drug development and what do we need to accelerate this? Could the authors also consider OA in animals - particularly horses and dogs - and whether joint chip development could be used to develop possible treatments or further our understanding of the disease in other species? Pregenerate (https://www.pregenerate.net/about) are developing cartilage chip models using animal cells, and the authors may want to be aware of their research.
Figure 1 is very clear and beautifully presented, I am not sure that the animals add anything as you do not mention these in the legend. I really like the colour coding.
Animals are now mentioned in the legend.
The section on Joint on chip is very good, I think that you should try and mirror this for the section on organoids, which is not as clear.
I also have these specific, minor issues that I think you could address:
Line 22: I think it would be more relevant to refer to more recent data regarding disease prevalence (eg Hunter et al. 2020 Osteoarthritis in 2020 and beyond: a Lancet Commission. The Lancet).
Referee 2 is right, citation was added and the sentence modified as follow:
“In 2019 OA affected about 7% of the world’s population.”
Lines 31-33 - I agree that 2D cultures are not sophisticated enough to model the joint, but this explanation is too generic and I think that you need to refer to the architecture and cell types of the joint, not just refer to "an organ", since your review is focused on the joint specifically, not organs in general.
Agree. The sentence was modified to specifically talk about the joint:
“However, due to their simplicity these models cannot properly recapitulate the complexity of an organ like joint such as intracellular signaling, fluid forces or influence of tissue interconnexion.”
Line 33-35: This seems misleading, animals can contribute to better understanding of animal biology, but can only lend assumptions to the understanding of human biology! We need to study humans to truly understand human biology.
Agree. The sentence was modified to avoid misleading:
“On the other hand, animal models bring this missing organ complexity and have largely contributed to better understand the function of the joint.”
Table 1 - in the first row, you need to explain what you mean by "interspecies" and there are also other issues associated with animals (costs, infrastructure, skills required, etc). For Cell culture - I do not think that you are correct- do you mean that cell cultures cannot recreate intercellular communication?
The table was corrected following instructions of referee 2.
Line 48- I think you need to insert the term "in vitro" after "microenvironments" for clarity
“In vitro” was added after microenvironments.
Line 48 - you need to clarify what you mean by "humanised models" -are you referring to in vitro/ex vivo models based on human cells or in vivo animals that are transgenic for human genes?
We were referring to in vitro/ex vivo models based on human cells.
For clarity, the sentence was modified:
“Development of new human joint biological models in order to predict the efficacy and toxicity of new treatments in rheumatological diseases is therefore essential while challenging.”
Line 57-59 is not clear- as I understand it, 3D printing does not always/automatically integrate fluid circulation etc, but 3D printed structures can be incorporated into dynamic models to produce the features that you describe.
Exactly, for clarity the sentence was modified too:
“Thanks to perfusion bioreactors for 3D constructs [10], those powerful tools can integrate fluid circulation, the delivery of growth factors & cytokines, and physical stimulation including oxygen level, external pressure, mobility.”
And a citation was added:
Gaspar, D.A.; Gomide, V.; Monteiro, F.J. The Role of Perfusion Bioreactors in Bone Tissue Engineering. Biomatter 2012, 2, 167–175, doi:10.4161/biom.22170
Line 84-91 This is an interesting example, did the authors of the paper (Broeren et al) create models with non-arthritic cells as a comparator? It is not clear what this research added to the field- the need for continued stimulation of cultures to maintain the pro-inflammatory profile, or the fact the these cells could be cultured together as a disease model?
This paper describe the use of RA synovial cells and show close interactions with monocytes, and contribute to the concept of arthritic synovial pannus as a model of disease.
Line 105-106 regarding the effects of celecoxib needs a reference.
A reference was added:
Tsutsumi, R.; Ito, H.; Hiramitsu, T.; Nishitani, K.; Akiyoshi, M.; Kitaori, T.; Yasuda, T.; Nakamura, T. Celecoxib Inhibits Production of MMP and NO via Down-Regulation of NF-ΚB and JNK in a PGE2 Independent Manner in Human Articular Chondrocytes. Rheumatol Int 2008, 28, 727–736, doi:10.1007/s00296-007-0511-6.
Line 110 - can you define where the microspheres were implanted- I assume this is an animal model but that needs to be clarified.
The microsphere were cultivated in vitro in adapted device. They can further be implanted in scid mice.
Line 114 It is Young's modulus (not Yong's)
Corrected.
Line 115-116, it would help to describe more about this model that you are developing - what results did you achieve when you altered the pressure and how does this mirror the disease? Can you comment on the possible development of this model for drug screening/understanding disease progression?
The model can be used to understand the physiopathological process (for example induced by repeated stimulation increase in pressure as well screen for new therapeutical strategies .
Line 140 Define the abbreviations FRZ-B and DKK1 please
FRZ-B, DKK1 and others abbreviations were written out the first time they are used.
Line 141 this sentence does not make sense
"different treatments already validated in human disease as osteoarthritis"
Sentence was modified:
“In this device, the authors were able to analyze the effect of validated treatments for osteo-arthritis such as a non-steroidal anti-inflammatory drug, dexamethasone, hyaluronic acid and the IL-1b antagonist and showed a beneficial effect on the expression of metalloproteases validating the device.”
Line 157 I am not sure what you mean by this:
"In order to reconstruct an articulation on a chip, we will rely on biomaterials, 3D culture and mini-bioreactors or microfluidic models available today." The English is not quite clear.
The sentence was also modified:
“Thus, in order to mimic in vitro an articulation, the biomechanical aspect must be taken into account. These mechanical stimulations can be provided by mini-bioreactors or microfluidic models available today.”
Line 167 Define MMP and ADAM please.
Abbreviations were written out the first time they are used.
Line 171 Define IPS
Abbreviations were written out the first time they are used.
Line 174 I disagree with the statement "3D bioprinting can overcome the organ-on-chip limitation to mimic the complexity of joint tissue." as I think you are oversimplifying both techniques! I think that 3D bioprinting offers some advantages, but it cannot substitute for OOC technologies and we need to consider combining this with other approaches, such as OOC, then you can start to build a more complex model with flow, vasculature, immune cell circulation etc., but I do not agree that these techniques are mutually exclusive, as you seem to be suggesting.
Referee 2 is right, so the sentence was modified in order to focus only on the size limitation of the OOC system:
“3D bioprinting can overcome the organ-on-chip size limitation to mimic the complexity of joint tissue.”
Line 201- could you clarify this please- do you mean for joints specifically or in general?
Sentence was modified:
“Indeed, a large number of studies aim at developing new suitable biomaterials for 3D bioprinting in general.”
Line 206: I do not know what you mean by "doping agents" and perhaps you could explain or rephrase?
Doping agents was replaced by bioactive molecules.
Round 2
Reviewer 1 Report
The authors have adequately addressed my concerns.
Reviewer 2 Report
The manuscript has been improved by revision and I thank the authors for their efforts, I would now recommend publication.